# Factors associated with clinically relevant pain reduction after a self-management program including education and exercise for people with knee and/or hip osteoarthritis: Data from the BOA register

**Thérése Jönsson**[1]*, **Frida Eek**[2], **Eva Ekvall Hansson**[2], **Leif E. Dahlberg**[3], **Andrea Dell'Isola**[4]

**1** Department of Health Sciences, Sport Sciences, Lund University, Lund, Sweden, **2** Department of Health Sciences, Human Movement: Health and Rehabilitation, Lund University, Lund, Sweden, **3** Department of Clinical Sciences Lund, Orthopedics, Lund University, Lund, Sweden, **4** Department of Clinical Sciences Lund, Clinical Epidemiology Unit, Orthopedics, Lund University, Lund, Sweden

\* therese.jonsson@med.lu.se

**Data Availability Statement:** A non-author point of contact (i.e., an email address) through which

## Abstract

### Aim

To examine the associations between individual- and disease-related factors and the odds of reaching a clinically relevant pain reduction in people with knee and/or hip osteoarthritis (OA) who underwent a first-line self-management program.

### Materials and methods

An observational registry-based study including people with knee (n = 18,871) and hip (n = 7,767) OA who participated in a self-management program including education and exercise and had data recorded in the Better Management of patients with Osteoarthritis (BOA) register. We used multivariable logistic regression models to study the association between sex, age, body mass index (BMI), education, comorbidity, pain frequency, walking difficulties, willingness to undergo surgery and the odds of reaching a clinically relevant pain reduction (decrease of >33% on a 0–10 NRS scale) 3 and 12 months after the intervention. All analyses were stratified by joint (knee/hip).

### Results

Both in the short- and long-term follow-up, a younger age (18–65 years), a lower BMI (< 25), a higher level of education (university), the absence of comorbidities impacting the ability to walk, less frequent pain and not being willing to undergo surgery were associated with higher odds of reaching a clinically relevant pain reduction in people with knee OA. We found similar results for people with hip OA, but with larger uncertainty in the estimates (wider 95% CI).

researchers can contact Lund University and request access to your data have been added: DHSdataaccess@med.lu.se.

**Funding:** This work was supported by Region Skåne (TJ) and Kockska foundation (TJ) https://www.skane.se/en/ http://www.kockskastiftelsen.se. The authors FE, EEH, LD and AD received no specific funding for this work. The funders had no role in the study design, data collection and analysis, decision to publish, or preparation of the manuscript. There was no additional external funding received for this study.

**Competing interests:** Therese Jönsson is the director of the Better Management of Patients with Osteoarthritis registry, for which she receives an honorarium. Leif Dahlberg is the co-founder and Chief Medical Officer of Joint Academy, a company which provides digital non-surgical treatment for patients with hip and knee osteoarthritis. Leif Dahlberg also owns stocks in, is a board member of, and is a paid consultant of Joint Academy. Joint Academy has otherwise not supported this work financially. There are no patent, products in development, or marketed products to declare. Frida Eek, Eva Ekvall Hansson, and Andrea Dell 'Isola has no competing interest to declare.

## Conclusion

Our study suggests that early fist line self-management interventions delivered when people have unilateral hip or knee OA with less frequent pain and are unwilling to undergo surgery, may be important for reaching a clinically relevant pain reduction after participation. Providing the most appropriate treatment to the right patient at the right time is a step in reducing the burden of OA for society and the patient.

## Introduction

Osteoarthritis (OA) in the knee and/or hip is the most common joint disease in the world and a high contributor to global disability [1]. In both knee and hip OA, pain is a cardinal symptom and typically become more severe, more frequent, and more disability over time significantly impacting a person quality of life and psychological wellbeing [1, 2]. In the absence of disease-modifying interventions, available first-line treatments including patient education and individualized exercise aim to reduce pain and improve function [3]. According to national and international guidelines, first-line core treatments should be offered to all people with OA of the knee and/or hip [4–7].

The Better Management of Patients with Osteoarthritis (BOA) is a Swedish National Quality register that evaluates the results from a self-management program following national and international guidelines and includes patient education and individual-adapted exercises, conducted at primary health care units all over Sweden [8]. Results from the BOA register has demonstrated improvement in pain at 3- and 12-months follow-up in people with knee and/or hip OA [9–11], but how different people respond to the self-management program in BOA, vary greatly [10, 11]. Previous studies have shown that factors like OA location (hip or knee), sex, age, body mass index (BMI), comorbidity, duration of symptoms, and pain at baseline are associated with outcome after a physiotherapy (PT) intervention in people with OA in the knee and/or hip [12–17]. However, most studies have shown an association between these factors and change in pain, but few studies have evaluated whether these associations are clinically relevant. Therefore, this study aimed to examine associations between individual- as well as disease-related factors and the odds of clinically relevant pain reduction after participation in a self-management program, delivered at a primary care level, in people with knee and/or hip OA.

## Materials and method

### Design and sample selection

The present study was an observational registry-based study and comprises data from the BOA register between 2008 and 2016. The BOA register contains data from people with knee and/or hip OA who have been participating in a self-management program including education and individual-adapted exercise [8]. The inclusion criteria for the participants to access the self-management program were symptoms from knee and/or hip that resulted in contact with the health care system. The exclusion criterions were, a reason other than OA for joint problems (e.g., sequel hip fractures, chronic widespread pain, inflammatory joint diseases, neuromuscular diseases or cancer); total joint replacement within the past 12 months; other surgery of the knee or hip joint within the past 3 months; and people not able to read or understand Swedish. The BOA register contains PT-reported data about the most affected joint,

previous treatment, and compliance to the intervention and patient-reported outcomes from participants in the self-management program [8]. To be included in the present study, the participants must have received at least the theory part of the self-management program and have data from baseline, 3- and 12-months follow-up. A two months' delay for the 3-months follow-up and a three months' delay for the 12-months follow-up were allowed for pragmatic reasons, based on an expected delay for some people due to unforeseen circumstances. Data about the index joint (knee or hip), were extracted from the physiotherapy form; in the case of bilateral or multi-joint OA, the most affected joint was chosen by the PT. The participants in the BOA register were treated in more than 500 different care units at a primary care level in Sweden.

### Intervention

The self-management program has previously been described [8, 11]. Briefly, it consists of a mandatory part with two theoretical group sessions led by a physiotherapist (PT) with 7–12 participants in each group. Following the education, participants can decide to undergo a face-to-face session with a trained PT who designs a personalized exercise program based on the participant's needs and goals. Patients could thereafter choose to perform the exercises on their own (at home), or during PT-supervised group exercise classes twice a week for 6–8 weeks. The intervention is followed up by an individual visit at 3 months and by a postal questionnaire at 12 months. All participants in the self-management program are supposed to fill in a questionnaire at baseline, 3- and 12-months follow-up.

### Ethics statement

The study was approved by the Regional Ethical Review Board in Gothenburg (1059–16).

The data we have used is manually registered in the BOA registry, which is a national quality register separate from patients' medical records. Use of data from a national quality register is regulated by the Swedish patient data act. To be registered in a national quality register, it is required that the patient is informed and given the opportunity to opt out. All people in present study have received oral or written information about the registration to the BOA register. The information must include that the data may be used for research after approval from a research ethical board. The research ethical board decides if consent is required or not, and if data should be anonymized. The ethical board decision was that no further information or consent is required, and that data must be anonymized, which is the case in present study.

### Measures

**Outcome.** *Clinical relevant pain reduction*. A Numeric Rating Scale (NRS pain 0–10) was used to record baseline pain intensity, asking for the mean pain during the last week [18, 19]. A clinically relevant pain reduction was defined as a decrease of >33% on NRS for pain. This cut-off is based on a prospective cohort study from Salaffi et al there they assessed patient's pain intensity by the numerical rating scale (NRS) at baseline and at the 3-month follow-up, and by a patient's global impression of change (PGIC) questionnaire [20]. A reduction of 33% was defined as a clinical relevant pain reduction and associated with feeling "much better" [20]. This cut-off has previously been validated in a sample of people with OA and other chronic rheumatic conditions [20, 21]. The decrease in NRS pain was calculated between baseline and 3- as well as 12-months follow-up.

**Independent/exposure factors.** *Individual factors*. Included independent variables regarding individual factors were sex (men/women), age categorized into three age groups: working age (18–64 years), younger retirees (65–74 years), and older retirees (≥75 years). BMI

was classified according to WHO into underweight ($<18.5$ kg/m$^2$), normal weight (18.5–24.9 kg/m$^2$), overweight (25–29.9 kg/m$^2$) and obese ($\geq 30$ kg/m$^2$) [22]. Because of low numbers in the underweight category, underweight and normal-weight people were merged into one category. Education level was divided into three groups: compulsory school, high school, and university.

*Disease-related factors*. The Charnley classification is a measure on the impact of comorbidity on walking score and categorizes people into one of three groups: A–one joint with osteoarthritis (unilateral knee or hip); B–bilateral osteoarthritis (both knees or both hips); C–osteoarthritis in multiple joint sites (hip and knee) or presence of any other disease that affects walking ability [23]. Pain frequency was assessed by the question: "How often do you have pain in your knee/hip," with five possible answers: never, every month, every week, every day, or all the time. Because of low numbers in the categories never and every month, the two were merged into one category. Walking disability was assessed by the question: "Do you have a walking disability caused by your OA" (Yes/No). Willingness to undergo surgery was assessed by the question: "Are your knee/hip symptoms so severe that you wish to undergo surgery?" (Yes/No).

*Covariates*. The covariates health-related quality of life (EQ-5D VAS), NRS pain at baseline, and previous surgery to the most affected joint were included in the analyses as potential confounders. Quality of life was measured using the EuroQol five dimensions visual analog scale (EQ-5D VAS 0–100) [24]. The EQ-5D VAS was used to adjust for baseline mental status as recommended by the International Consortium for Health Outcomes Measurement for hip and/or knee OA [25]. Previous surgery to the most affected joint was recorded by the PT at baseline, this question was used to adjust for worst symptoms due to surgery; studies indicate that previous surgery is associated longitudinally with worse symptoms in people with knee OA [26].

## Statistics

All statistical analyses were performed using IBM SPSS Statistics (version 25.0, IBM Corp., Armonk, NY, USA). All the analyses were stratified based on the most affected joint (knee, or hip). Descriptive statistics were conducted to provide an overview of the specific characteristics of the participants in the study. Crude (separate for each independent/exposure variable) and multivariable logistic regression models were applied to examine the odds of reaching a clinically relevant pain reduction at 3- and 12-months follow-up for groups based on individual- and disease-related factors. The results are presented using odds ratios (ORs) and 95% confidence intervals (CIs) from both crude and adjusted models. The level of significance was set at $p < 0.05$.

## Results

A total of 51,627 people (mean age 66 years, 70% female) with knee or hip OA participated in the mandatory theory part and were eligible for the study. Of these, 26,638 people with the knee (n = 18,871) and hip (n = 7,767) had data from at least one of the follow-ups and were included in the study. The baseline characteristics of included and excluded people with knee and hip OA and the response rate are described in Tables 1 and 2. The reason for dropouts is described in Fig 1. At the 3-month follow-up, 43% of people with knee OA and 37% of people with hip OA experienced clinically relevant pain reduction. At the 12-month follow up 38% of people with knee OA and 29% of people with hip OA reach clinically relevant pain reduction.

**Table 1. Characteristics of patients with knee OA population (n = 18871).**

| Variables | Knee | | | | | |
|---|---|---|---|---|---|---|
| | | | *3-month follow-up* | | *12-month follow-up* | |
| | | | *Pain reduction* | | *Pain reduction* | |
| | **Excluded** | **Total group** | **≥ 33%** | **< 33%** | **≥ 33%** | **< 33%** |
| | **n = 1 6497** | **n = 1 8871** | **n = 8 197** | **n = 1 0674** | **n = 7 161** | **n = 11 495** |
| Sex, % (n) | | | | | | |
| women | 69 (11 424) | 70 (13 297) | 71 (5 802) | 70 (7 495) | 71 (5 111) | 70 (8 027) |
| Missing, *n* | *44* | *0* | *0* | *0* | *0* | *0* |
| Age, % (n) | | | | | | |
| 18–64 | 43 (7 111) | 39 (7 272) | 39 (3 210) | 38 (4 062) | 40 (2 833) | 38 (4 372) |
| 65–74 | 40 (6 562) | 43 (8 208) | 44 (3 635) | 43 (4 573) | 45 (3 196) | 43 (4 903) |
| > = 75 | 17 (2 780) | 18 (3 391) | 17 (1 352) | 19 (2 039) | 16 (1 132) | 19 (2 220) |
| Missing, *n* | *44* | *0* | *0* | *0* | *0* | *0* |
| Body mass index, % (n) | | | | | | |
| < 25 | 22 (3 666) | 26 (4 808) | 28 (2 228) | 25 (2 580) | 29 (2 054) | 24 (2 692) |
| ≥25–30 | 42 (6 951) | 44 (8 171) | 45 (3 601) | 44 (4 570) | 45 (3 161) | 44 (4 920) |
| >30 | 33 (5 470) | 30 (5 571) | 28 (2 233) | 32 (3 338) | 26 (1 839) | 33 (3 672) |
| Missing, *n* | *410* | *321* | *135* | *186* | *107* | *211* |
| Education, % (n) | | | | | | |
| Compulsory school | 35 (5 666) | 34 (6 346) | 32 (2 612) | 35 (3 734) | 30 (2 129) | 36 (4 124) |
| High school | 38 (6 268) | 37 (6 914) | 37 (2 982) | 37 (3 932) | 37 (2 607) | 37 (4 232) |
| University | 27 (4 447) | 29 (5 560) | 32 (2 582) | 28 (2 978) | 34 (2 410) | 27 (3 103) |
| Missing, *n* | *116* | *51* | *21* | *30* | *15* | *36* |
| Charnley Category*, % (n) | | | | | | |
| A | 37 (6 099) | 39 (7 360) | 45 (3 661) | 35 (3 699) | 48 (3 439) | 33 (3 828) |
| B | 22 (3 661) | 24 (4 502) | 23 (1 890) | 25 (2 612) | 22 (1 594) | 25 (2 856) |
| C | 40 (6 672) | 37 (6 986) | 32 (2 642) | 41 (4 344) | 30 (2 122) | 42 (4 794) |
| Missing, *n* | *65* | *23* | *4* | *19* | *6* | *17* |
| NRS pain** (0–10), mean (SD) | 5.4 (2) | 5.1 (1.9) | 5.4 (1.8) | 5 (2) | 5.3 (1.9) | 5 (2) |
| Missing, *n* | *109* | *0* | *0* | *0* | *0* | *0* |
| Pain frequency, % (n) | | | | | | |
| Less than every month | 7 (1 155) | 6 (1 215) | 7 (551) | 6 (660) | 7 (518) | 6 (674) |
| Every week | 13 (2 145) | 13 (2 455) | 14 (1 132) | 12 (1 312) | 14 (1 033) | 12 (1 383) |
| Every day | 61 (1 0063) | 62 (11 840) | 64 (5 245) | 61 (6 536) | 64 (4 549) | 62 (7 109) |
| All the time | 19 (3 134) | 18 (3 391) | 15 (1 251) | 20 (2 130) | 15 (1 043) | 20 (2 294) |
| Missing | *49* | *63* | *18* | *36* | *18* | *35* |
| Walking difficulties % (n) | | | | | | |
| Yes | 79 (13 032) | 78 (14 787) | 78 (6 371) | 78 (8 346) | 76 (5 425) | 80 (9 122) |
| Missing, *n* | *164* | *107* | *40* | *61* | *40* | *59* |
| Previous surgery, % (n) | | | | | | |
| Yes | 19 (3 061) | 17 (3 206) | 16 (1 318) | 18 (1 888) | 15 (1.070) | 18 (2103) |
| Missing, *n* | *46* | *44* | *12* | *20* | *13* | *19* |
| EQ-%D VAS*** (0–100), mean (SD) | 66 (19) | 68 (19) | 70 (18) | 67 (19) | 71 (18) | 67 (19) |
| Missing, *n* | *2 520* | *3 698* | *1 488* | *2 210* | *1 306* | *2 308* |
| Willingness of surgery, % (n) | | | | | | |
| Yes | 27 (4 480) | 20 (3 777) | 17 (1 378) | 22 (2 319) | 15 (1 073) | 23 (2 570) |

(*Continued*)

**Table 1.** (Continued)

| Variables | | | Knee | | | |
|---|---|---|---|---|---|---|
| | | | 3-month follow-up | | 12-month follow-up | |
| | | | Pain reduction | | Pain reduction | |
| | Excluded | Total group | ≥ 33% | < 33% | ≥ 33% | < 33% |
| | n = 1 6497 | n = 1 8871 | n = 8 197 | n = 1 0674 | n = 7 161 | n = 11 495 |
| Missing, n | 247 | 160 | 63 | 97 | 57 | 100 |

\* Charnley Category A, one joint with OA (unilateral knee or hip); B, bilateral OA (both knees or both hips); C, OA in multiple joint sites (hip and knee), or presence of any other disease that affects walking ability,

\*\* Numeric rating scale,

\*\*\* EuroQol five-dimensional visual analogue scale.

### Individual and disease-related factors

**Individual factors.** Adjusted models showed that people with knee OA with younger age, lower BMI, and a higher level of education were more likely to reach a clinically relevant pain reduction both in the short and long term (Table 3). In people with hip OA, adjusted models showed that people with a lower BMI were more likely to reach clinically relevant pain reduction at the 12-month follow-up (Table 4). A younger age decreased the odds to reach clinically relevant pain reduction at the 12-month follow-up for people with hip OA (Table 4).

**Disease-related factors.** Adjusted models showed that people with knee OA and/or hip OA with Charnley A and Charnley B, less frequent pain and not having the willingness to undergo surgery were more likely to reach a clinically relevant pain reduction both at the three and 12-month follow-up (Tables 3 and 4). People with no walking difficulties were more likely to reach clinically relevant pain reduction at the 12-month follow-up (Tables 3 and 4).

## Discussion

This study aimed to explore the association of individual and disease-related factors and clinically relevant pain reduction after participation in a self-management program for people with knee or hip OA. The results showed that those with unilateral OA, less frequent pain, and unwillingness to undergo surgery were more likely to reach clinically relevant pain reduction in both the short and long term.

The result from the present study suggests that participation in a self-management program for OA early in the disease course may increase the patient's odds to reach a clinically relevant pain reduction. An increased pain frequency or intermittent pain has previously been shown to be correlated with unacceptable symptoms for patients with OA [27]. In the present study, people with pain frequency less than every month had higher odds to reach a clinically relevant pain reduction than people with pain all the time. Furthermore, the results indicate a trend towards increasing odds to reach clinically relevant pain reduction with decreasing pain frequency (Tables 3, 4). OA is a chronic disease that requires continuous treatment accompanied by behavioral changes. Reaching a clinically relevant pain reduction early in the disease may, therefore, foster an active approach and facilitate the behavioral changes necessary for the long-term self-management of OA symptoms [28]. Furthermore, it seems to be more important to reach the patients early in the disease course than in younger age. In the present study, we included all the people who participated in the self-management program which can be accessed by any person with OA from an age of 18 years. Patients younger than 65 years with knee OA had higher odds to reach a clinically relevant pain reduction at 12-month follow-up

**Table 2. Characteristics of patients with hip OA (n = 7767).**

| Variables | Hip | | | | | |
|---|---|---|---|---|---|---|
| | | | 3-month follow-up | | 12-month follow-up | |
| | | | *Pain reduction* | | *Pain reduction* | |
| | **Excluded** | **Total group** | **≥ 33%** | **< 33%** | **≥ 33%** | **< 33%** |
| | **n = 8 399** | **n = 7 767** | **n = 2 862** | **n = 4 868** | **n = 2 259** | **n = 5 363** |
| Sex, % (n) | | | | | | |
| women | 67 (5 616) | 70 (5 462) | 71 (2 036) | 70 (3 396) | 71 (1 610) | 70 (3 756) |
| Missing, *n* | *18* | *0* | *0* | *0* | *0* | *0* |
| Age, % (n) | | | | | | |
| 18–64 | 37 (3 117) | 33 (2 573) | 33 (957) | 33 (1 609) | 32 (722) | 34 (1 809) |
| 65–74 | 43 (3 603) | 46 (3 605) | 48 (1 373) | 46 (2 216) | 49 (1 099) | 46 (2 447) |
| > = 75 | 20 (1 661) | 21 (1 589) | 19 (532) | 21 (1 043) | 19 (438) | 21 (1 107) |
| Missing, *n* | *18* | *0* | *0* | *0* | *0* | *0* |
| Body mass index, % (n) | | | | | | |
| < 25 | 32 (2 688) | 37 (2 834) | 38 (1 083) | 36 (1 738) | 41 (918) | 35 (1 862) |
| ≥25–30 | 43 (3 643) | 42 (3 267) | 43 (1 216) | 43 (2 039) | 43 (962) | 43 (2 249) |
| >30 | 23 (8 225) | 20 (1 546) | 19 (527) | 21 (1 010) | 16 (349) | 22 (1 168) |
| Missing, *n* | *174* | *120* | *36* | *81* | *30* | *84* |
| Education, % (n) | | | | | | |
| Compulsory school | 35 (2 946) | 35 (2 687) | 34 (956) | 35 (1 714) | 33 (740) | 35 (1 880) |
| High school | 37 (3 072) | 35 (2 697) | 34 (960) | 36 (1 728) | 34 (754) | 36 (1 897) |
| University | 28 (2 331) | 30 (2 355) | 33 (937) | 29 (1 407) | 34 (760) | 29 (1 563) |
| Missing, *n* | *50* | *28* | *9* | *19* | *5* | *23* |
| Charnley Category*, % (n) | | | | | | |
| A | 38 (3 186) | 38 (2 911) | 39 (1 143) | 36 (1 753) | 43 (967) | 35 (1 878) |
| B | 9 (766) | 12 (896) | 12 (347) | 11 (546) | 13 (289) | 11 (596) |
| C | 53 (4 420) | 51 (3 949) | 48 (1 366) | 53 (2 564) | 44 (1 001) | 54 (2 880) |
| Missing, *n* | *27* | *11* | *6* | *5* | *2* | *9* |
| NRS pain** (0–10), mean (SD) | 5.6 (1.9) | 5.1 (1.9) | 5.4 (1.8) | 5 (2) | 5.4 (1.8) | 5.1 (2) |
| Missing, *n* | *45* | *50* | *0* | *0* | *0* | *0* |
| Pain frequency, % (n) | | | | | | |
| Less than every month | 4 (297) | 5 (416) | 6 (172) | 5 (241) | 6 (130) | 5 (279) |
| Every week | 9 (792) | 13 (1 023) | 14 (405) | 13 (612) | 15 (344) | 12 (662) |
| Every day | 63 (5 273) | 64 (4 947) | 65 (1 845) | 63 (3 081) | 65 (1 471) | 63 (3 398) |
| All the time | 24 (1 999) | 17 (1 350) | 15 (425) | 19 (920) | 14 (305) | 19 (1 004) |
| Missing | *38* | *31* | *14* | *15* | *9* | *20* |
| Walking difficulties % (n) | | | | | | |
| Yes | 85 (7 165) | 78 (6 022) | 76 (2 177) | 79 (3 822) | 73 (1 658) | 79 (4 244) |
| Missing, *n* | *66* | *44* | *21* | *22* | *14* | *29* |
| Previous surgery, % (n) | | | | | | |
| Yes | 2 (172) | 2 (135) | 2 (45) | 2 (88) | 1 (32) | 2 (102) |
| Missing, *n* | *21* | *20* | *9* | *10* | *4* | *16* |
| EQ-5D VAS*** (0–100), mean (SD) | 66 (19) | 69 (19) | 69 (18) | 66 (19) | 69 (18) | 66 (19) |
| Missing, *n* | *1294* | *1429* | *492* | *937* | *404* | *1081* |
| Willingness of surgery, % (n) | | | | | | |
| Yes | 36 (3 005) | 20 (3 713) | 16 (463) | 23 (1 096) | 15 (325) | 22 (1 184) |

(*Continued*)

**Table 2.** (Continued)

| Variables | Hip | | | | | |
|---|---|---|---|---|---|---|
| | | | 3-month follow-up | | 12-month follow-up | |
| | | | Pain reduction | | Pain reduction | |
| | Excluded | Total group | ≥ 33% | < 33% | ≥ 33% | < 33% |
| | n = 8 399 | n = 7 767 | n = 2 862 | n = 4 868 | n = 2 259 | n = 5 363 |
| Missing, n | 109 | 175 | 18 | 45 | 16 | 48 |

* Charnley Category A, one joint with OA (unilateral knee or hip); B, bilateral OA (both knees or both hips); C, OA in multiple joint sites (hip and knee), or presence of any other disease that affects walking ability,

** Numeric rating scale,

*** EuroQol five-dimensional visual analogue scale.

OR (95% CI), 1.24 (1.11-1-.37), while the same pattern was not found among patients with hip OA. One reason for the difference may be due to the higher prevalence of hip OA due to abnormal anatomy or hip diseases in childhood among younger patients. The presence of such abnormalities may potentially reduce the benefit associated with exercise and physical activity, partially explaining the observed results. Nevertheless, we could not verify the presence of such abnormalities in our sample.

In the present study, people with unilateral OA (Charnley A) and bilateral OA without other comorbidities (Charnley B) were more likely to reach clinically relevant pain reduction to the self-management program than people with OA in multiple joint sites (hip and knee), or presence of any other disease that affects walking ability (Charnley C). The association between comorbidities and response is in line with previous studies [12, 14, 15]. A high number of comorbidities may restrict a patient's ability to participate in certain parts of the program, which may affect the result of the intervention. One treatment does not fit all people and we probably need to individualize the treatment better for people with comorbidities.

To be unwilling to undergo surgery was associated with higher odds to reach a clinically relevant pain reduction. People who are willing to undergo surgery before starting the first-line intervention may see surgery as the best solution for their problem and may have lower expectations and hence lower motivation for a self-management program [29]. Furthermore, people seeking surgical management may have more severe symptoms as suggested by a recently published study using BOA data which an association between willingness to undergo surgery and higher pain intensity [30]. Finally, it must be considered that in some regions in Sweden it is mandatory to participate in a first-line treatment program before surgery. While it is important to maximize the number of people receiving education and exercise before undergoing surgical interventions, the lack of a shared decision-making process behind the choice of the

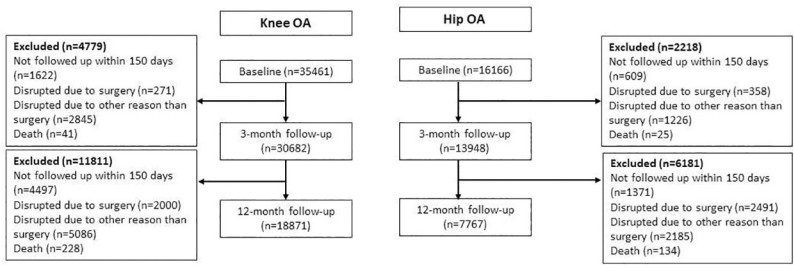

**Fig 1. Flow-chart of the study population.**

**Table 3. Factors associated with reaching a clinically relevant pain reduction at 3- and 12-month follow-up in people with knee OA.**

| Independent factors | Knee | | | | | |
| --- | --- | --- | --- | --- | --- | --- |
| | *3-month follow-up* | | | *12-month follow-up* | | |
| | *≥ 33% pain reduction* | | | *≥ 33% pain reduction* | | |
| | n | Crude | Adjusted | n | Crude | Adjusted |
| | | OR (95% CI) | OR (95% CI) | | OR (95% CI) | OR (95% CI) |
| Sex | | | | | | |
| women | 10 178 | 0.97 (0.91–1.04) | 0.98 (0.91–1.06) | 10 085 | **1.08 (1.01–1.15)** | 0.97 (0.9–1.05) |
| men | 4 442 | 1 | 1 | 4 407 | 1 | 1 |
| Age | | | | | | |
| 18–64 | *5 471* | **1.19 (1.1–1.3)** | **1.18 (1.06–1.3)** | 5 436 | **1.27 (1.17–1.39)** | **1.24 (1.11–1.37)** |
| 65–74 | *6 441* | **1.2 (1.1–1.3)** | **1.16 (1.05–1.23)** | 6 375 | **1.28 (1.18–1.39)** | **1.21 (1.09–1.33)** |
| > = 75 | *2 704* | 1 | 1 | | 1 | 1 |
| Body mass index | | | | | | |
| < 25 | *3 859* | **1.29 (1.2–1.4)** | **1.28 (1.17–1.41)** | 3 822 | **1.52 (1.41–1.65)** | **1.38 (1.25–1.52)** |
| ≥25–30 | *6 417* | **1.18 (1.1–1.26)** | **1.13 (1.04–1.23)** | 6 361 | **1.28 (1.19–1.38)** | **1.23 (1.13–1.43)** |
| >30 | *4 340* | 1 | 1 | 1 | 1 | 1 |
| Education | | | | | | |
| Compulsory school | *4 775* | *1* | 1 | *4 721* | 1 | 1 |
| High school | *5 473* | **1.08 (1.01–1.16)** | 1.05 (0.97–1.15) | *5 426* | **1.19 (1.11–1.28)** | **1.16 (1.07–1.27)** |
| University | *4 369* | **1.24 (1.15–1.33)** | **1.2 (1.11–1.32)** | *4 345* | **1.5 (1.4–1.62)** | **1.45 (1.33–1.59)** |
| Charnley Category* | | | | | | |
| A | 5 934 | **1.63 (1.52–1.74)** | **1.62 (1.5–1.76)** | 5 878 | **2.03 (1.9–2.17)** | **1.96 (1.8–2.1)** |
| B | 3 069 | **1.19 (1.1–1.28)** | **1.14 (1.04–1.26)** | 3 045 | **1.26 (1.16–1.37)** | **1.2 (1.08–1.3)** |
| C | 5 613 | 1 | 1 | 5 569 | 1 | 1 |
| Pain frequency | | | | | | |
| Less than every month | 981 | **1.42 (1.25–1.62)** | **2.18 (1.8–2.6)** | 968 | **1.69 (1.48–1.94)** | **2.12 (1.78–2.54)** |
| Every week | 1 914 | **1.47 (1.32–1.63)** | **1.9 (1.7–2.18)** | 1 893 | **1.64 (1.47–1.83)** | **1.82 (1.58–2.09)** |
| Every day | 9 198 | **1.37 (1.26–1.48)** | **1.46 (1.33–1.61)** | 9 127 | **1.41 (1.3–1.53)** | **1.42 (1.29–1.58)** |
| All the time | 2 523 | 1 | 1 | 2 504 | 1 | 1 |
| Walking difficulties | | | | | | |
| No | 3 208 | 1.03 (0.96–1.1) | 1.0 (0.92–1.1) | 3177 | **1.23 (1.14–1.3)** | **1.14 (1.04–1.25)** |
| Yes | 11 408 | 1 | 1 | 11 315 | 1 | 1 |
| Willingness of surgery | | | | | | |
| No | 11 521 | **1.38 (1.3–1.48)** | **1.6 (1.4–1.7)** | 12 125 | **1.64 (1.5–1.77)** | **1.7 (1.54–1.88)** |
| Yes | 2 781 | 1 | 1 | 2 367 | 1 | 1 |

* Charnley Category A, one joint with OA (unilateral knee or hip); B, bilateral OA (both knees or both hips); C, OA in multiple joint sites (hip and knee), or presence of any other disease that affects walking ability. All adjusted analyses are adjusted for all variables in the model including health-related quality of life (EQ-5D-VAS), baseline pain (NRS pain), and previous surgery to the knee. Bold text is an increased or decreased odds ratio (p<0.05).

treatment has the potential to negatively impact the results of an intervention [29]. This may partially explain why people willing to undergo surgery had lower odds of response in our study.

The results from the present study indicate that it seems important to undergo a first-line self-management program early in the disease course, as long the people only have unilateral OA, less frequent pain, and do not have a willingness of surgery. Currently, only 50% of patients with OA receive the treatments as recommended by existing guidelines [31]. Reports from many different countries have shown similar situations suggesting an overall lack of

**Table 4. Factors associated with reaching a clinically relevant pain reduction at 3- and 12-month follow-up in people with hip OA.**

| Independent factors | Hip | | | | | |
| --- | --- | --- | --- | --- | --- | --- |
| | 3-month follow-up | | | 12-month follow-up | | |
| | Pain reduction | | | ≥ 33% pain reduction | | |
| | n | Crude | Adjusted | n | Crude | Adjusted |
| | | OR (95% CI) | OR (95% CI) | | OR (95% CI) | OR (95% CI) |
| Sex | | | | | | |
| women | 4 279 | 1.07 (0.97–1.18) | 0.97 (0.86–1.09) | 4 244 | 1.06 (0.95–1.18) | 0.94 (0.83–1.07) |
| men | 1 831 | 1 | 1 | 1 811 | 1 | 1 |
| Age | | | | | | |
| 18–64 | 1 949 | **1.17 (1.02–1.33)** | 1.15 (0.98–1.34) | 1 932 | 1.01 (0.88–1.16) | **0.96 (0.81–0.98)** |
| 65–74 | 2 871 | **1.22 (1.07–1.38)** | 1.11 (0.97–1.28) | 2 853 | 1.14 (0.99–1.3) | 1.02 (0.88–1.19) |
| > = 75 | 1 290 | 1 | 1 | 1 | 1 | 1 |
| Body mass index | | | | | | |
| < 25 | 2 255 | **1.19 (1.05–1.36)** | 1.11 (0.96–1.29) | 2 236 | **1.65 (1.43–1.9)** | **1.5 (1.28–1.78)** |
| ≥25–30 | 2 578 | **1.14 (1.01–1.3)** | 1.06 (0.92–1.23) | 2 555 | **1.43 (1.24–1.65)** | **1.35 (1.15–1.58)** |
| >30 | | 1 | 1 | 1.264 | 1 | 1 |
| Education | | | | | | |
| Compulsory school | 2 086 | 1 | 1 | 2 058 | 1 | 1 |
| High school | 2 153 | 0.99 (0.89–1.11) | 0.95 (0.83–1.09) | 2 132 | 1.01 (0.89–1.14) | 0.99 (0.86–1.15) |
| University | 1 871 | **1.19 (1.07–1.34)** | 1.14 (0.99–1.31) | 1 865 | **1.23 (1.09–1.39)** | 1.15 (0.99–1.33) |
| Charnley Category* | | | | | | |
| A | 2 244 | **1.22 (1.1–1.35)** | **1.23 (1.1–1.39)** | 2 219 | **1.48 (1.33–1.65)** | **1.44 (1.27–1.63)** |
| B | 529 | **1.19 (1.03–1.39)** | **1.24 (1.02–1.5)** | 527 | **1.39 (1.19–1.63)** | **1.29 (1.23–1.32)** |
| C | 3 337 | 1 | 1 | 3 309 | 1 | 1 |
| Pain frequency | | | | | | |
| Less than every month | 343 | **1.55 (1.23–1.94)** | **2.39 (1.76–3.1)** | 338 | **1.53 (1.2–1.96)** | **2.2 (1.63–2.98)** |
| Every week | 828 | **1.43 (1.21–1.7)** | **1.72 (1.39–2.13)** | 823 | **1.71 (1.43–2.05)** | **1.9 (1.51–2.39)** |
| Every day | 3 902 | **1.29 (1.14–1.47)** | **1.39 (1.19–1.62)** | 3 873 | **1.43 (1.24–1.64)** | **1.46 (1.23–1.73)** |
| All the time | 1 037 | 1 | 1 | 1 019 | 1 | 1 |
| Walking difficulties | | | | | | |
| No | 1 339 | **1.14 (1.02–1.27)** | 1.11(0.97–1.28) | 1331 | **1.38 (1.23–1.55)** | **1.32 (1.15–1.53)** |
| Yes | 4 771 | 1 | 1 | 4724 | 1 | 1 |
| Willingness of surgery | | | | | | |
| No | 4 880 | **1.5 (1.34–1.71)** | **1.75 (1.5–2.04)** | 4 842 | **1.69 (1.48–1.93)** | **1.86 (1.57–2.2)** |
| Yes | 1 230 | 1 | 1 | 1 213 | 1 | 1 |

* Charnley Category A, one joint with OA (unilateral knee or hip); B, bilateral OA (both knees or both hips); C, OA in multiple joint sites (hip and knee), or presence of any other disease that affects walking ability. All adjusted analyses are adjusted for all variables in the model including health-related quality of life (EQ-5D-VAS), baseline pain (NRS pain), and previous surgery to the knee. Bold text is an increased or decreased odds ratio (p<0.05).

implementation of first-line treatments. This in turn indicates the importance of continuing to work on the implementation of guidelines for OA to help to reduce the rising burden of OA.

## Strengths and limitations of the study

This study has important strengths as it investigates the association of patients' characteristics and response to a first-line intervention in a large sample of more than 26 000 patients with knee and hip OA. Moreover, randomized controlled trials often apply stringent inclusion criteria which may result in study samples that do not reflect the OA population seeking care.

The intervention analyzed in the study is provided nationwide in primary care settings and therefore reflects closely current clinical practices in real-world settings. This study includes "real-world" data which increases the generalizability of the results even if only to people attending the self-management program. Some limitations need to be addressed. The number of excluded people at 3- and 12-months follow-up was high, as expected in real-world settings for registry-based studies. This limits the generalizability of the results only to people attending the follow-ups after participating in a self-management program. However, the baseline characteristics of the excluded people only differed from those included in the study regarding the willingness to undergo surgery (Tables 1 and 2). In the present study, a previously suggested cut-off on improvement in NRS pain, corresponding to "much better", was used to define the outcome of clinically relevant pain reduction [20]. When measuring outcomes on a continuous scale as it is routinely done for pain, the "regression-to-the-mean effect" can occur [32]. At the extreme ends of the scale, people can only change in one direction. Therefore, people with baseline scores far above the average show higher improvements in change scores than people with low pain. As expected, we also observed a difference in pain intensity at baseline, where people who reached a clinically relevant pain reduction had on average higher baseline pain than the people who did not reach a clinically relevant pain reduction. We attempted to adjust for the regression-to-the-mean effect by adjusting for baseline pain, and by using the percentage change instead of the change of score to define responders. Thus, we believe that these results are relevant and may help to identify people who are likely to reach a clinically relevant pain reduction following education and individual-adapted exercise both in the short and long term. Finally, due to the observational nature of the study, our results should not be interpreted as inferring causality.

## Conclusions

In this study, we showed that people with unilateral OA, less frequent pain, and unwillingness to undergo surgery were more likely to reach clinically relevant pain reduction after participating in a self-management program including education and exercise directed to people with knee and/or hip OA. Furthermore, people with a lower BMI were more likely to respond in the long term. Providing the most appropriate treatment to the right patient at the right time is a step in reducing the burden of OA for society and the patient.

## Acknowledgments

The authors would like to acknowledge all participating patients, PTs reporting data to the BOA registry, and others involved in the BOA registry.

## Author Contributions

**Formal analysis:** Thérése Jönsson, Frida Eek, Andrea Dell'Isola.

**Investigation:** Thérése Jönsson.

**Methodology:** Thérése Jönsson, Frida Eek, Andrea Dell'Isola.

**Supervision:** Eva Ekvall Hansson, Leif E. Dahlberg, Andrea Dell'Isola.

**Writing – original draft:** Thérése Jönsson.

**Writing – review & editing:** Thérése Jönsson, Frida Eek, Eva Ekvall Hansson, Leif E. Dahlberg, Andrea Dell'Isola.

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
