## [Decision Letter · Decision Letter 0]

9 Sep 2022

PONE-D-21-40195Factors associated with clinically relevant pain reduction after a self-management program including education and exercise for people with knee and/or hip osteoarthritis: data from the BOA registerPLOS ONE

Dear Dr. Jönsson,

Thank you for submitting your manuscript to PLOS ONE. After careful consideration, we feel that it has merit but does not fully meet PLOS ONE’s publication criteria as it currently stands. Therefore, we invite you to submit a revised version of the manuscript that addresses the points raised during the review process.

Interesting study with  large sample size  and outcomes.

1- It might be better to explain  more and elaborate  about how to   estimate and  use a cut-off 33 percent 

2- I would suggest to  consider another  age group  ( younger )  in category  however it  might not affect the results but  methodologically would be sound  and robust

3- Please  elaborate  co-existing diseases however  Charnley score mentioned  but  needs more explanation  about what excatlly   included and how  to calculate

4- Please draw separate tables for Knee and Hip  for better presentation  and make it easier to look for reader

5- I would suggest to consult with an expert in biostatics regarding to use Linear  Mixed Model  instead of Logistic  with considering the  uncertainly in determining the cut off  and also some variability in 500 centers however this is only a suggestion

We look forward to receiving your revised manuscript.

Kind regards,

Hamid Reza Baradaran, M.D., Ph.D.,

Academic Editor

PLOS ONE

Journal Requirements:

“This work was supported by Region Skåne (TJ) and Kockska foundation (TJ). https://www.skane.se/en/
http://www.kockskastiftelsen.se .The funders had no role in study design, data collection and analysis, decision to publish, or preparation of the manuscript.”

4. Thank you for stating the following in the Competing Interests/Financial Disclosuresection:

“Thérése Jönsson is a member of the steering committee of the Swedish National Quality Registry Better Management of patients with Osteoarthritis (The BOA registry), for which she does not receive any compensation. Leif Dahlberg is the co-founder and Chief Medical Officer of Joint Academy, a company which provides digital non-surgical treatment for patients with hip and knee osteoarthritis. Leif Dahlberg also owns stocks in, is a board member of, and is a paid consultant of Joint Academy. Joint Academy has otherwise not supported this work financially. There are no patents, products in development, or marketed products to declare.”

We note that one or more of the authors are employed by a commercial company: name of commercial company.

5. We noted in your submission details that a portion of your manuscript may have been presented or published elsewhere. [Fig 1 has been published in PLOS ONE one time before by Therese Jönsson (me). The figure is describing the intervention used in this article and will make it easier for the reader. The article has been attached by the files namned "Related Manuscript filetype] Please clarify whether this [conference proceeding or publication] was peer-reviewed and formally published. If this work was previously peer-reviewed and published, in the cover letter please provide the reason that this work does not constitute dual publication and should be included in the current manuscript.

Additional Editor Comments (if provided):

Interesting study with large sample size and outcomes. I

1- It might be better to explain more and elaborate about how to estimate and use a cut-off 33 percent

2- I would suggest to consider another age group ( younger ) in category however it might not affect the results but methodologically would be sound and robust

3- Please elaborate co-existing diseases however Charnley score mentioned but needs more explanation about what excatlly included and how to calculate

4- Please draw separate tables for Knee and Hip for better presentation and make it easier to look for reader

5- I would suggest to consult with an expert in biostatics regarding to use Linear Mixed Model instead of Logistic with considering the uncertainly in determining the cut off and also some variability in 500 centers however this is only a suggestion

Reviewers' comments:

Reviewer's Responses to Questions

**Comments to the Author**

1. Is the manuscript technically sound, and do the data support the conclusions?

Reviewer #1: Yes

2. Has the statistical analysis been performed appropriately and rigorously? 

Reviewer #1: Yes

3. Have the authors made all data underlying the findings in their manuscript fully available?

Reviewer #1: Yes

4. Is the manuscript presented in an intelligible fashion and written in standard English?

Reviewer #1: Yes

5. Review Comments to the Author

Reviewer #1: PONE-D-21-40195

Thank you for the opportunity to review the manuscript entitled “Factors associated with clinically relevant pain reduction after a self-management”.

• The study is well-designed and provides findings to support the appropriate treatment to the right patient at the right time; however, there are some issues with the writing quality that need to be addressed before publication acceptance.

• The study's main strengths are the sample size and completion of follow-up, though it is unclear whether any patients withdraw from the study after 12 months. Furthermore, the inclusion criteria for the study were not carefully chosen.

• It would have been preferable, in my opinion, if patients with knee or hip osteoarthritis had been included in the study. Why did the researchers choose to study both diseases?

• Authors are advised to list the inclusion criteria in detail including pain intensity, …

• Were data from patients who had an evaluation included in the study?

• It seems that the exclusion criteria should be more than the mentioned items, such as neuromuscular disease, bone implants; history of knee intra-articular injection in the past three months, …

• Fig. 1 does not provide enough detailed information about how to conduct and evaluate the study. This figure does not include the number of patients enrolled in the study or the possible withdrawal of some of them.

• Why did you consider the age range of osteoarthritis patients to be so broad, beginning at the age of 18?

• Given that primary osteoarthritis begins in middle age, wouldn't it be better, in my opinion, to consider the age range beginning around 40 years? Then, the samples were more homogeneous in this study, making the study results more reliable.

• Because people with osteoarthritis participate in this study, and because the majority of sufferers seek treatment and rehabilitation programs due to pain, there is no need to include "never" in the Pain frequency category.

• The number of patients who were excluded from the study is unknown.

• The authors did not investigate the reasons for the relationships revealed between the study variables in the discussion.

• The findings of this study can be guessed to some extent based on the results of previous studies; however, the authors should emphasize the study's strengths more.

6. PLOS authors have the option to publish the peer review history of their article (what does this mean?). If published, this will include your full peer review and any attached files.

Reviewer #1: No

---

## [Author Response · Author response to Decision Letter 0]

8 Dec 2022

Dear Editor,

Thank you for your response and for the reviewers’ constructive comments and suggestions providing us with the opportunity to improve our manuscript. Changes have been made in order to meet the criticism raised, and in the few cases where we have not made the suggested changes, our reasons for not doing so are given. 

We hereby submit the revised version of the paper along with the response to each of the specific comments provided by you and the reviewers.

EDITOR'S SPECIFIC COMMENT

Interesting study with large sample size and outcomes. 

1. Editor comment - It might be better to explain more and elaborate about how to estimate and use a cut-off 33 percent

Author response – Thank you to get us the opportunity to explain our choice of cut-off for clinically relevant pain reduction. As all of us are aware of, it is hard to define a cut-off for clinically relevant pain reduction in patients with osteoarthritis. The reference we used, was a prospective cohort study assessing a patient's pain intensity by the numerical rating scale (NRS) at baseline and the 3-month follow-up, and by a patient's global impression of change (PGIC) questionnaire. A one-unit difference at the lowest end of the PGIC (“slightly better”) was used to define MCID as it reflects the minimum and lowest degree of improvement that could be detected. In addition, they also calculated the NRS changes best associated with “much better”. To characterize the association between specific NRS change scores (raw or per cent) and clinically important improvement, the sensitivity and specificity were calculated by the receiver operating characteristic (ROC) method. They used PGIC as an external criterion to distinguish between improved or non-improved patients. Their results showed that on average, a reduction of one point or a reduction of 15.0% in the NRS represented an MCID for the patient. A change in the NRS score of −2.0 and a per cent change score of −33.0% were best associated with the patients reporting feeling “much better” [1]. For this reason, these values can be considered appropriate cut-off points for this measure. 

Author action – To motivate our choice of cut-off for clinically relevant pain reduction, the following sentence has been added: lines 110-115, This cut-off is based on a prospective cohort study from Salaffi et al there they assessed patient's pain intensity by the numerical rating scale (NRS) at baseline and at the 3-month follow-up, and by a patient's global impression of change (PGIC) questionnaire [20]. This cut-off is based on a prospective cohort study from Salaffi et al there they assessed patient's pain intensity by the numerical rating scale (NRS) at baseline and at the 3-month follow-up, and by a patient's global impression of change (PGIC) questionnaire [20]. A reduction of 33% was defined as a clinical relevant pain reduction and associated with feeling “much better” [20].

2. Editor comment - I would suggest considering another age group ( younger ) in the category however it might not affect the results but methodologically would be sound and robust.

Author response - Although it is not common for people around 18 years old with OA, people with OA from 18 years old are included in the BOA registry. We agree with the reviewer that exploring this age group may be interesting and information on younger age groups is seldomly reported. In our study, we are interested to describe the whole population who attend the supported self-management program in Sweden. Unfortunately, the proportion of younger persons (ie <40 years) in our sample is small (n < 180). Considering the small number of patients in this age group, we would like to refrain to perform the analysis with this added subgroup which we expect will not provide meaningful results due to the potentially very large 95% CIs that will accompaign the estimates. 

Author action – The following sentences have been added to the discussion part, line 271-281. Furthermore, it seems to be more important to reach the patients early in the disease course than in younger age. In the present study, we included all the people who participated in the self-management program which can be accessed by any person with OA from an age of 18 years. Patients younger than 65 years with knee OA had higher odds to reach a clinically relevant pain reduction at 12-month follow-up OR (95% CI), 1.24 (1.11-1-.37), while the same pattern was not found among patients with hip OA. One reason for the difference may be due to the higher prevalence of hip OA due to abnormal anatomy or hip diseases in childhood among younger patients. The presence of such abnormalities may potentially reduce the beefit associated with exercise and physical activity, partially explaining the observed results. Nevertheless we could not verify the presence of such abnormalities in our sample.

3. Editor comment - Please elaborate on co-existing diseases however Charnley score mentioned but needs more explanation about what excatlly included and how to calculate.

Author response – Thank you for letting us explain what the Charnley score includes. Charnley score is a measure of the impact of comorbidities on walking. It categorizes people with OA into 3 classes based on the disease(s) that affect walking ability (categorical variable – Charnley Score; A = unilateral hip or knee OA / B = bilateral hip or knee OA / C = multiple joint OA or some other condition). Unfortunately, we do not have any more data about comorbidities. In our analysis we have used Charnley A/B/C and evaluated the association with reaching a clinically relevant pain reduction, so no other calculations have been done. 

Author action – Line 126-130 have been changed to: The Charnley classification is a measure on the impact of comorbidity on walking score and categorizes people into one of three groups: A – one joint with osteoarthritis (unilateral knee or hip); B – bilateral osteoarthritis (both knees or both hips); C – osteoarthritis in multiple joint sites (hip and knee) or presence of any other disease that affects walking ability [23].

4. Editor comment - Please draw separate tables for Knee and Hip for better presentation and make it easier to look for reader.

Author response – Thank you for this comment. We agree with you that it will be easier to read the tables if we separate them into 2 tables.

Author action – Table 1 has been dived to Table 1, line 166 and Table 2, line 174. 

5. Editor comment - I would suggest consulting with an expert in biostatics regarding to use Linear Mixed Model instead of Logistic with considering the uncertainly in determining the cut off and also some variability in 500 centers however this is only a suggestion.

Author response - Thank you for your suggestion. We partly agree with your concern, since we are aware that the use of continuous variables is generally considered advantageous. However, in our study, we had the aim to examine the relation to “clinically relevant” improvements, based on a defined cut-offs for clinically relevant improvements. Although it is not to be considered a “black or white” change in the pain, this cutoff has been defined, evaluated and supported in previous studies. We also consider the interpretation of “odds of a clinically relevant improvement” to be more clinically interpretable. Also, we are aware that some variability between centres can be expected. Nevertheless, the self-management programme provided in the clinics is standardized which should considerably reduce the between-centres variability. Thus we don’t expect systematic differences between centres large enough that would require some form of “cluster adjustment”. We would therefore appreciate if we could keep our previously defined dichotomous outcome, and hence also the logistic regression model as we do consider that it responds to our aim with the study. 

Author action - We kept our binary outcome, but as previously mentioned we added a more clear motivation for the cut-off, line 110-115, This cut-off is based on a prospective cohort study from Salaffi et al there they assessed patient's pain intensity by the numerical rating scale (NRS) at baseline and at the 3-month follow-up, and by a patient's global impression of change (PGIC) questionnaire [20]. A reduction of 33% was defined as a clinical relevant pain reduction and associated with feeling “much better” [20].

REVIEWERS SPECIFIC COMMENTS:

Thank you for the opportunity to review the manuscript entitled “Factors associated with clinically relevant pain reduction after a self-management”. The study is well-designed and provides findings to support the appropriate treatment to the right patient at the right time; however, there are some issues with the writing quality that need to be addressed before publication acceptance.

1. Reviewers comment - The study's main strengths are the sample size and completion of follow-up, though it is unclear whether any patients withdraw from the study after 12 months. 

Author response – Thank you for highlighting this. Since we have used “real-world” data it is very common with missing follow-ups. However, we have added a flowchart to make the data transparent for all. 

Author action – The reason for dropouts is described in Fig 1, line 165.

Fig 1. Flowchart of the study population

2. Reviewers comment - Furthermore, the inclusion criteria for the study were not carefully chosen.

Author response – Thank you for letting us clarify the inclusion/exclusion criteria in this study. The patients included in this study underwent a standardized self-management intervention for OA which is provided nationwide in Sweden and for which the inclusion criteria are pre-specified and based on national and international guidelines (e.g. EULAR, NICE) for the management of OA, this means that it was not possible for us to modify the inclusion criteria to the intervention, line 82-88. For what concerns the criteria used to include patients in the study they were described in, line 74-81, “To be included in the present study, the participants must have received at least the theory part of the self-management program and have data from baseline, 3- and 12-months follow-up. A two months delay for the 3-months follow-up and a three months delay for the 12-month follow-up were allowed for pragmatic reasons, based on an expected delay for some people due to unforeseen circumstances. Data about the index joint (knee or hip), were extracted from the physiotherapy form; in the case of bilateral or multi-joint OA, the most affected joint was chosen by the PT”. These criteria were selected to be as inclusive as possible while minimizing the risk of selection bias that can arise from the use of more stringent criteria. However, we agree with the reviewer that more clarity regarding our choice of inclusion criteria may further improve our manuscript.

Author action – Line 68-74, “People with OA were allowed to access the self-management program if they had symptoms from knee and/or hip that resulted in contact with the health care system. People were not allowed to access the program if, there was a reason other than OA for joint problems (e.g., sequel hip fractures, chronic widespread pain, inflammatory joint diseases, neuromuscular diseases or cancer); they received total joint replacement within the past 12 months; other surgery of the knee or hip joint within the past 3 months; if they were not able to read or understand Swedish”.

Line: 74-77, “The BOA register contains PT-reported data about the most affected joint, previous treatment, and compliance to the intervention and patient-reported outcomes from participants in the self-management program [8]. To be included in the present study, the participants must have received at least the theory part of the self-management program and have data from baseline, 3- and 12-month follow-up. A two months delay for the 3-month follow-up and a three months delay for the 12-month follow-up were allowed for pragmatic reasons, based on an expected delay for some people due to unforeseen circumstances. Data about the index joint (knee or hip), were extracted from the physiotherapy form; in the case of bilateral or multi-joint OA, the most affected joint was chosen by the PT”.

3. Reviewer's comment - It would have been preferable, in my opinion, if patients with knee or hip osteoarthritis had been included in the study. Why did the researchers choose to study both diseases?

Author response – Thank you for the comments, we can see both positive and negative aspects to including both patients with knee and hip OA. However, the supported self-management program with patient education and exercise is usually conducted with both patients with knee and hip OA together, so we believe that it is interesting to include both. We also believe it is interesting to compare and believe that is easier if all data is in the same article. However, to make the manuscript easier to read we have made 2 tables instead of 1 to describe patients with knee or hip separately. 

Author action - None

4. Reviewers comment - Authors are advised to list the inclusion criteria in detail including pain intensity, …

Author response – We thank the reviewer for the comment. The inclusion in the self-management programme analysed in this study is not based on the level of pain but on the presence of a clinical diagnosis of OA which does not require a specific pain intensity [2]. Nevertheless, based on the reviewer previous comment we have provided clarifications to the inclusion criteria which we hope will answer all the reviewer’s concerns.

Author action – Please see our response to the reviewer’s comment number 2.

5. Reviewers comment - Were data from patients who had an evaluation included in the study?

Author response – We thank the reviewer for the comment. To be included in this study the patients needed to have the following inclusion criteria (as described in line 77-79), “To be included in the present study, the participants must have received at least the theory part of the self-management program and have data from baseline, 3- and 12-months follow-up”. 

Author action - None

6. Reviewers’ comment - It seems that the exclusion criteria should be more than the mentioned items, such as neuromuscular disease, bone implants; history of knee intra-articular injection in the past three months, …

Author response – We thank the reviewer for the comment, we have provided some examples of 

diseases to further help the reader in understanding the exclusion criteria. Nevertheless, we cannot provide a comprehensive list of conditions as the choice to include the patients is ultimately left to the clinicians assessing the patient at baseline. 

Author action – At line 72, neuromuscular diseases has been added to the list: “ The inclusion criteria for the participants to access the self-management program were symptoms from knee and/or hip that resulted in contact with the health care system. The exclusion criterions were, a reason other than OA for joint problems (e.g., sequel hip fractures, chronic widespread pain, inflammatory joint diseases, neuromuscular diaseases or cancer); total joint replacement within the past 12 months; other surgery of the knee or hip joint within the past 3 months; and people not able to read or understand Swedish.”, line 68-74.

7. Reviewers comment - Fig. 1 does not provide enough detailed information about how to conduct and evaluate the study. This figure does not include the number of patients enrolled in the study or the possible withdrawal of some of them.

Author response – We thank the reviewer for the comment, Fig 1 has been removed and replaced with a flowchart of the study to better understand the number of patients enrolled in the study and the reason for withdrawal.

Author action – At line 165, Fig 1. Flowchart of the study population has been added. 

8. Reviewers comment - Why did you consider the age range of osteoarthritis patients to be so broad, beginning at the age of 18? Given that primary osteoarthritis begins in middle age, wouldn't it be better, in my opinion, to consider the age range beginning around 40 years? Then, the samples were more homogeneous in this study, making the study results more reliable.

Author response – We thank the reviwer for the comment. Although it is not common for people around 18 years old with OA, people with OA from 18 years old are included in the BOA registry. We agree with the reviewer that exploring this age group may be interesting and information on younger age groups is seldomly reported. In our study, we are interested to describe the whole population who attend the supported self-management program in Sweden. Unfortunately, the proportion of younger persons (ie <40 years) in our sample is small (n < 180). Considering the small number of patients in this age group, we would like to refrain to perform the analysis with this added subgroup which we expect will not provide meaningful results due to the potentially very large 95% CIs that will accompaign the estimates. 

Author action - The following sentences have been added to the discussion part, line 271-281. Furthermore, it seems to be more important to reach the patients early in the disease course than in younger age. In the present study, we included all the people who participated in the self-management program which can be accessed by any person with OA from an age of 18 years. Patients younger than 65 years with knee OA had higher odds to reach a clinically relevant pain reduction at 12-month follow-up OR (95% CI), 1.24 (1.11-1-.37), while the same pattern was not found among patients with hip OA. One reason for the difference may be due to the higher prevalence of hip OA due to abnormal anatomy or hip diseases in childhood among younger patients. The presence of such abnormalities may potentially reduce the beefit associated with exercise and physical activity, partially explaining the observed results. Nevertheless we could not verify the presence of such abnormalities in our sample.

9. Reviewers comment - Because people with osteoarthritis participate in this study, and because the majority of sufferers seek treatment and rehabilitation programs due to pain, there is no need to include "never" in the Pain frequency category.

Author response – Thank you for your comment. We agree with you that pain is the cardinal symptom and the most common cause of patients seeking health care, however in the BOA registry, a question about pain frequency is taken from the KOOS and described in the manuscripts in lines 130-133: “Pain frequency was assessed by the question: “How often do you have pain in your knee/hip,” with five possible answers: never, every month, every week, every day, or all the time. Because of low numbers in the categories never and every month, the two were merged into one category”. Thus, we decided to maintain the category to reflect the formulation included in the questionnaire. 

Author action - None

10. Reviewers comment - The number of patients who were excluded from the study is unknown.

Author response – We thank the reviewer for the comment. We agree that more clarity regarding the exclusion of participants is needed. We have thus added a flowchart to clarify the selection process.

Author action: At line 165, Fig 1. Flowchart of the study population has been added.

11. Reviewers comment - The authors did not investigate the reasons for the relationships revealed between the study variables in the discussion.

Author response: We thank the reviewer for the comment. We agree that investigating the reason for the association between some of the included variables and the response to the treatment is of great interest. Nevertheless, this would not be possible with the current study design which was planned to answer a different research question and did not aim to (and cannot) explain causality. Nevertheless, we discussed possible hypotheses behind the observed association. However, we tried to keep these speculations to a minimum since high-quality data supporting a causal link behind the observed association is lacking.

Author action – No further action has been taken.

12. Reviewers comment - The findings of this study can be guessed to some extent based on the results of previous studies; however, the authors should emphasize the study's strengths more.

Author response – We agree with the reviewer that clearly stating the strength of this study will further improve our manuscript.

Author action – Line 311-318, “This study has important strengths as it investigates the association of patients’ characteristics and response to a first-line intervention in a large sample of more than 26 000 patients with knee and hip OA. Moreover, randomized controlled trials often apply stringent inclusion criteria which may result in study samples that do not reflect the OA population seeking care. The intervention analyzed in the study is provided nationwide in primary care settings and therefore reflects closely current clinical practices in real-world settings. This study includes “real-world” data which increases the generalizability of the results even if only to people attending the self-management program.”

1. Salaffi F, Stancati A, Silvestri CA, Ciapetti A, Grassi W: Minimal clinically important changes in chronic musculoskeletal pain intensity measured on a numerical rating scale. Eur J Pain 2004, 8(4):283-291.

2. Nationella riktlinjer för rörelseorganens sjukdomar 2012 [Elektroniska resurser]: osteoporos, artros, inflammatorisk ryggsjukdom och ankyloserande spondylit, psoriasisartrit och reumatoid artrit : stöd för styrning och ledning. Stockholm: Socialstyrelsen; 2012.

---

## [Editor Report · Decision Letter 1]

9 Feb 2023

Factors associated with clinically relevant pain reduction after a self-management program including education and exercise for people with knee and/or hip osteoarthritis: data from the BOA register

PONE-D-21-40195R1

Dear Dr. Jönsson,

We’re pleased to inform you that your manuscript has been judged scientifically suitable for publication and will be formally accepted for publication once it meets all outstanding technical requirements.

Kind regards,

Hamid Reza Baradaran, M.D., Ph.D.,

Academic Editor

PLOS ONE
---

## [Editor Report · Acceptance letter]

14 Feb 2023

PONE-D-21-40195R1 

Factors associated with clinically relevant pain reduction after a self-management program including education and exercise for people with knee and/or hip osteoarthritis: data from the BOA register 

Dear Dr. Jönsson:

I'm pleased to inform you that your manuscript has been deemed suitable for publication in PLOS ONE. Congratulations! Your manuscript is now with our production department. 

Kind regards, 

on behalf of

Professor Hamid Reza Baradaran 

Academic Editor

PLOS ONE